# Discovery of New *Trichophyton* Members, *T. persicum* and *T. spiraliforme* spp. nov., as a Cause of Highly Inflammatory Tinea Cases in Iran and Czechia

Adéla Čmoková,[a,b] Ali Rezaei-Matehkolaei,[c] Ivana Kuklová,[d] Miroslav Kolařík,[b] Forough Shamsizadeh,[e] Saham Ansari,[f] Maral Gharaghani,[e] Viera Miňovská,[d] Mohammad Javad Najafzadeh,[g] Sadegh Nouripour-Sisakht,[h] Takashi Yaguchi,[i] Kamiar Zomorodian,[j] Hossein Zarrinfar,[k] Vit Hubka[a,b,i]

[a]Department of Botany, Faculty of Science, Charles University, Prague, Czech Republic
[b]Laboratory of Fungal Genetics and Metabolism, Institute of Microbiology, Czech Academy of Sciences, Prague, Czech Republic
[c]Cellular and Molecular Research Center, Medical Basic Sciences Research Institute, Ahvaz Jundishapur University of Medical Sciences, Ahvaz, Iran
[d]Department of Dermatology and Venereology, First Faculty of Medicine, Charles University and General University Hospital in Prague, Prague, Czech Republic
[e]Infectious and Tropical Diseases Research Center, Health Research Institute, Ahvaz Jundishapur University of Medical Sciences, Ahvaz, Iran
[f]Department of Medical Parasitology and Mycology, School of Medicine, Shahid Beheshti University of Medical Sciences, Tehran, Iran
[g]Department of Parasitology and Mycology, Faculty of Medicine, Mashhad University of Medical Sciences, Mashhad, Iran
[h]Medicinal Plants Research Center, Yasuj University of Medical Sciences, Yasuj, Iran
[i]Medical Mycology Research Center, Chiba University, Chiba, Japan
[j]Department of Medical Mycology and Parasitology, School of Medicine, Shiraz University of Medical Sciences, Shiraz, Iran
[k]Allergy Research Center, Mashhad University of Medical Sciences, Mashhad, Iran

**ABSTRACT** Pathogens from the *Trichophyton benhamiae* complex are one of the most important causes of animal mycoses with significant zoonotic potential. In light of the recently revised taxonomy of this complex, we retrospectively identified 38 *Trichophyton* isolates that could not be resolved into any of the existing species. These strains were isolated from Iranian and Czech patients during molecular epidemiological surveys on dermatophytosis and were predominantly associated with highly inflammatory tinea corporis cases, suggesting possible zoonotic etiology. Subsequent phylogenetic (4 markers), population genetic (10 markers), and phenotypic analyses supported recognition of two novel species. The first species, *Trichophyton persicum* sp. nov., was identified in 36 cases of human dermatophytosis and one case of feline dermatophytosis, mainly in Southern and Western Iran. The second species, *Trichophyton spiraliforme* sp. nov., is only known from a single case of tinea corporis in a Czech patient who probably contracted the infection from a dog. Although the zoonotic sources of infections summarized in this study are very likely, little is known about the host spectrum of these pathogens. Awareness of these new pathogens among clinicians should refine our knowledge about their poorly explored geographic distribution.

**IMPORTANCE** In this study, we describe two novel agents of dermatophytosis and summarize the clinical manifestation of infections. These new pathogens were discovered thanks to long-term molecular epidemiological studies conducted in Czechia and Iran. Zoonotic origins of the human infections are highly probable, but the animal hosts of these pathogens are poorly known. Further research is needed to refine our knowledge about these new dermatophytes.

**KEYWORDS** dermatophytosis, molecular epidemiology, multigene phylogeny, skin mycoses, *Trichophyton benhamiae* complex, zoonotic infections, zoophilic dermatophytes

**Citation** Čmoková A, Rezaei-Matehkolaei A, Kuklová I, Kolařík M, Shamsizadeh F, Ansari S, Gharaghani M, Miňovská V, Najafzadeh MJ, Nouripour-Sisakht S, Yaguchi T, Zomorodian K, Zarrinfar H, Hubka V. 2021. Discovery of new *Trichophyton* members, *T. persicum* and *T. spiraliforme* spp. nov., as a cause of highly inflammatory tinea cases in Iran and Czechia. Microbiol Spectr 9:e00284-21. https://doi.org/10.1128/Spectrum.00284-21.

Address correspondence to Vit Hubka, vit.hubka@gmail.com.

Dermatophytes are one of the most common pathogens of animals and humans worldwide (1). Because dermatophytosis is not a life-threatening infection and its health impact can be relatively low, it represents a rather neglected infection in many

regions. The infections with mild or moderate symptoms and chronic course are mostly caused by anthropophilic dermatophytes, which are typically localized to body areas that have higher moisture levels or are less accessible for the host immune response, e.g., skin folds, feet, and nails (2–4). On the other hand, zoophilic dermatophytes, when transmitted to a human, usually cause highly inflammatory infections located in sites directly exposed to contact with an infected animal, typically the extremities, face, and trunk (5, 6). These differences are usually attributed to adaptation of pathogens to their primary hosts leading to a balanced immune response, i.e., a mild immune response is usually associated with infections caused by anthropophilic species which coevolved with humans, and the opposite is true for zoophilic species (7–9).

A significant portion of the zoophilic *Trichophyton* species belongs to the *Trichophyton benhamiae* complex, which encompasses nine taxa. These are mostly animal pathogens whose hosts include pets, livestock, and free-living animals (10). The only clearly anthropophilic species in the complex is *Trichophyton concentricum*, a cause of tinea imbricata (Tokelau) in rural indigenous populations in the tropics (11, 12). In the past, *Trichophyton verrucosum* was the best-known and most studied species from this complex. It is a well-known agent of dermatophytosis in cattle and of zoonotic infections, especially in agricultural workers (13, 14). But with the introduction of new agricultural procedures and vaccination of cattle, it has been almost eradicated in many regions (15, 16). While the number of dermatophytoses due to *T. verrucosum* in Europe decreased significantly in the last decades, the interest of clinicians has been aroused by the emergence of infections due to *T. benhamiae*, whose number has increased significantly, especially in various European countries but also in Japan (17–20). In humans, *T. benhamiae* is usually the causative agent of tinea corporis and tinea capitis, transmitted mainly from guinea pigs but also from many other minor hosts, such as dogs (mostly reported from North America), rabbits, and various rodents. It was shown that *T. benhamiae* is a complex of several zoophilic species which are partly different in their geographic distribution and host range (10). Apart from *T. benhamiae sensu lato*, the complex includes another emerging pathogen that is associated with hedgehogs, *Trichophyton erinacei*, which is increasingly reported worldwide due to the growing interest of people in pet hedgehogs (21, 22).

The rise of cultivation techniques at the end of the 19th century represented a boom in describing new dermatophyte species, followed by another, smaller wave due to mating experiments in the 1960s and 1970s (23). Huge molecular studies at the turn of the millennium brought rather a decline in the number of accepted species (24, 25). Since then, only seven novel *Trichophyton* members have been described, three of which, however, belong to the geophilic genus *Arthroderma* based on the current taxonomy (26–30). Thus, novel primary pathogenic *Trichophyton* species are restricted only to *Trichophyton indotineae* (31), a member of the *Trichophyton mentagrophytes* complex, and three novel zoophilic species belonging to the *T. benhamiae* complex, namely, *Trichophyton europaeum*, *Trichophyton japonicum*, and *Trichophyton africanum* (10).

Thousands of strains were collected during epidemiological studies in Iran and Czechia, and their identification verified based on DNA sequencing. This approach led to the discovery of two new species belonging to the *T. benhamiae* complex that are proposed in this study, using a polyphasic approach comprising morphology, physiology, and molecular data.

## RESULTS

**Tinea cases due to *Trichophyton persicum*.** In total, 37 cases of dermatophytosis caused by *T. persicum* were identified during retrospective analysis of internal transcribed spacer (ITS), ribosomal DNA (rDNA), and *tef1-α* sequences generated during epidemiological surveys in Iran conducted between 2008 and 2019. One case of feline dermatophytosis was also identified. The cases are summarized in Table 1. Originally, the *T. persicum* isolates were identified by Iranian medical mycologists as *T. verrucosum* or *Microsporum canis*, based on morphology. Anamnestic data about the contact of patients with animals were usually missing. The only exception was the case of a 31-year-old woman (isolate S366)

**TABLE 1** Overview of tinea cases attributed to the newly described *Trichophyton* species

| Species, strain[a] | Region or country | Locality | Sex[b] | Age (yr) | Clinical manifestation | Date(s) of isolation | GenBank accession no. | | | |
|---|---|---|---|---|---|---|---|---|---|---|
| | | | | | | | ITS rDNA | tef1-α | gapdh | tubb |
| *Trichophyton persicum* | | | | | | | | | | |
| Ahv-376 | Southwestern Iran | Ahvaz | F | 34 | Tinea corporis | 2013 | KT192449 | | | |
| Ahv-200 | Southwestern Iran | Ahvaz | F | 68 | Tinea corporis | 2013 | KT192448 | | | |
| Ahv-419 | Southwestern Iran | Ahvaz | F | 23 | Tinea corporis | 2013 | KT192450 | | | |
| 237 | Southwestern Iran | Ahvaz | F | 26 | Tinea corporis | 2012–2014 | MW936601 | | | |
| 302 | Southwestern Iran | Ahvaz | F | 54 | Tinea corporis | 2012–2014 | MW936603 | | | |
| RM6 | Central Iran | Isfahan | F | 45 | Tinea corporis | 2011 | MW936621 | | | |
| M6 | Northeastern Iran | Mashhad | F | 20 | Tinea corporis | 2013 | MF850250 | | | |
| CCF 6543[T] (mums91B, MNJ-91) | Eastern Iran | Qaen | F | 5 | Tinea capitis (ectothrix) | 2017 | MW936609 | MG356864 | MW959142 | MW959139 |
| MNJ-104 | Eastern Iran | Qaen | M | 14 | Tinea corporis | 2017 | MW936612 | | | |
| MNJ-120P (mums120B) | Eastern Iran | Qaen | M | 10 | Tinea corporis | 2017 | MW936613 | MG356866 | | |
| MNJ-154 (mums154b) | Eastern Iran | Qaen | M | 45 | Tinea corporis | 2017 | MW936614 | MG356865 | | |
| MNJ-155 (mums155B) | Eastern Iran | Qaen | M | 12 | Tinea corporis | 2017 | MW936615 | MG356867 | | |
| MNJ-16 (mums16B) | Eastern Iran | Qaen | F | 6 | Tinea corporis | 2017 | MW936616 | MG356860 | | |
| MNJ-28 (mums28B) | Eastern Iran | Qaen | F | 35 | Tinea corporis | 2017 | MW936617 | MG356861 | | |
| MNJ-32 (mums32B) | Eastern Iran | Qaen | M | 9 | Tinea corporis | 2017 | MW936618 | MG356862 | | |
| MNJ-6761 | Eastern Iran | Qaen | F | 29 | Tinea corporis | 2017 | MW936619 | | | |
| MNJ-81 (mums81B) | Eastern Iran | Qaen | M | 5 | Tinea corporis | 2017 | MW936620 | MG356863 | | |
| 5 | Southern Iran | Shiraz | M | 26 | Tinea corporis | 2012–2014 | MW936604 | | | |
| 25 | Southern Iran | Shiraz | F | 31 | Tinea corporis | 2012–2014 | MW936602 | | | |
| 58 | Southern Iran | Shiraz | F | 24 | Tinea corporis | 2012–2014 | MW936607 | | | |
| 72 | Southern Iran | Shiraz | F | 38 | Tinea corporis | 2012–2014 | MW936608 | | | |
| 172 | Southern Iran | Shiraz | F | 34 | Tinea corporis | 2012–2014 | MW936599 | | | |
| 174 | Southern Iran | Shiraz | F | 48 | Tinea corporis | 2012–2014 | MW936600 | | | |
| T286 | Southern Iran | Shiraz | F | 23 | Tinea corporis | 2017–2019 | MW936624 | | | |
| T289 | Southern Iran | Shiraz | F | 55 | Tinea corporis | 2017–2019 | MW936625 | | | |
| T290 | Southern Iran | Shiraz | F | 14 | Tinea corporis | 2017–2019 | MW936626 | | | |
| T321 | Southern Iran | Shiraz | F | 13 | Tinea corporis | 2017–2019 | MW936627 | | | |
| T322 | Southern Iran | Shiraz | M | 10 | Tinea corporis | 2017–2019 | MN807357 | | | |
| Ci1947 | Northern Iran | Tehran | M | 24 | Tinea corporis | 2012 | MW936610 | | | |
| SH1 | Northern Iran | Tehran | M | 25 | Tinea corporis | 2016 | MW936623 | | | |
| KH-M4 | Northwestern Iran | Meshkin-shahr | — | — | Dermatophytosis in a stray cat | 2015 | MW936611 | | | |
| Ahv-523 | Southwestern Iran | Yasuj | F | 16 | Tinea corporis | 2013 | KT192451 | | | |
| Ahv-525 | Southwestern Iran | Yasuj | M | 2 | Tinea corporis | 2013 | KT192452 | | | |
| S366 | Southwestern Iran | Yasuj | F | 31 | Tinea corporis | 2013 | MN808768 | | | |
| S394 | Southwestern Iran | Yasuj | F | 16 | Tinea corporis | 2013 | MW936622 | | | |
| 501 | Southwestern Iran | Yasuj | F | 45 | Tinea corporis | 2012–2014 | MW936605 | | | |
| 502 | Southwestern Iran | Yasuj | M | 10 | Tinea corporis | 2012–2014 | MW936606 | | | |
| *Trichophyton spiraliforme* | | | | | | | | | | |
| CCF 6259[T] | Czech Republic | Hostouň | M | 43 | Tinea corporis | 2017 | MW936628 | MW959141 | MW959143 | MW959140 |

[a]Superscript T, ex-type strain.
[b]F, female; M, male.

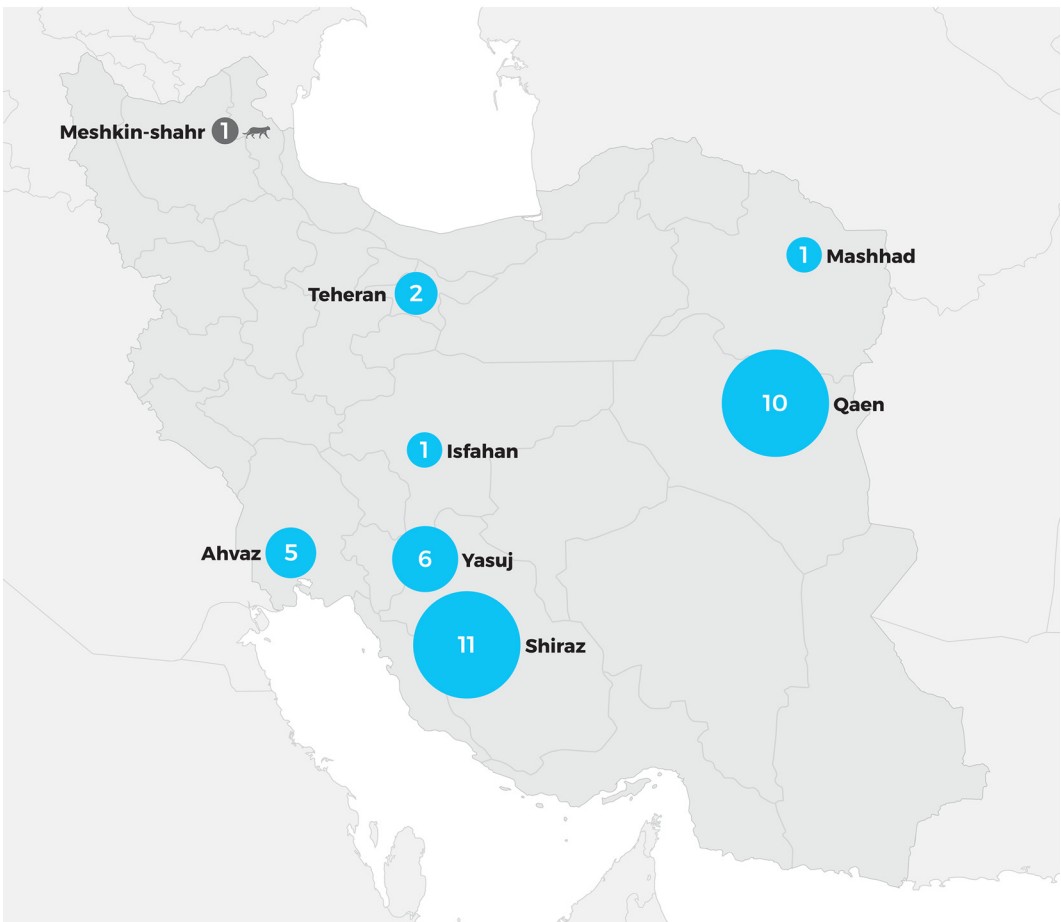

**FIG 1** Geographic map distribution of tinea cases identified as due to *Trichophyton persicum* in Iran.

who lived in a rural area of Iran and reported contact with various farm animals, including cats and cattle.

In general, the cases of dermatophytosis due to *T. persicum* were reported especially from the area of Southern, Southwestern, and Eastern Iran (Fig. 1). The first case of dermatophytosis caused by *T. persicum*, from 2011, was traced in Isfahan, followed by 17 cases that originated mainly in Southern and Southwestern Iran (Shiraz, Yasuj, and Ahvaz) during 2012 to 2014. The last and larger group of cases occurred in the area of Eastern Iran (10 cases in Qaen) during 2017, together with a few cases traced mainly in Shiraz during 2017 to 2019.

Almost all infections due to *T. persicum* manifested as tinea corporis (Table 1), except for a patient with a scalp infection (ectothrix). The exact body area of infections was usually missing in the anamnestic data, but in the cases with known details, the lesions were located mainly on the upper extremities (dorsum of hand, arm, elbow, and wrist) and, less frequently, on the trunk, face, and feet (Fig. 2A to C). The youngest patient was only 2 years old and the oldest 68 years old. Even though the patients' age range is wide, young adults (median age 24) were the most frequently affected. Females represented two-thirds of patients, with a median age of 30 years, while the median age of male patients was 11 years.

**Tinea corporis due to *Trichophyton spiraliforme*.** A 43-year-old immunocompetent male patient presented with a painful inflammatory skin change on the inner aspect of his left knee. There was no history of prior trauma to the site. He was otherwise healthy except for a history of congenital hyperbilirubinemia and penicillin allergy. The patient was first treated with roxithromycin 150 mg twice a day orally and ammonium bituminosulfonate ointment (Ichtoxyl ung) topically based on the initial diagnosis of bacterial pyoderma. There was no visible improvement at the 10-day follow-up.

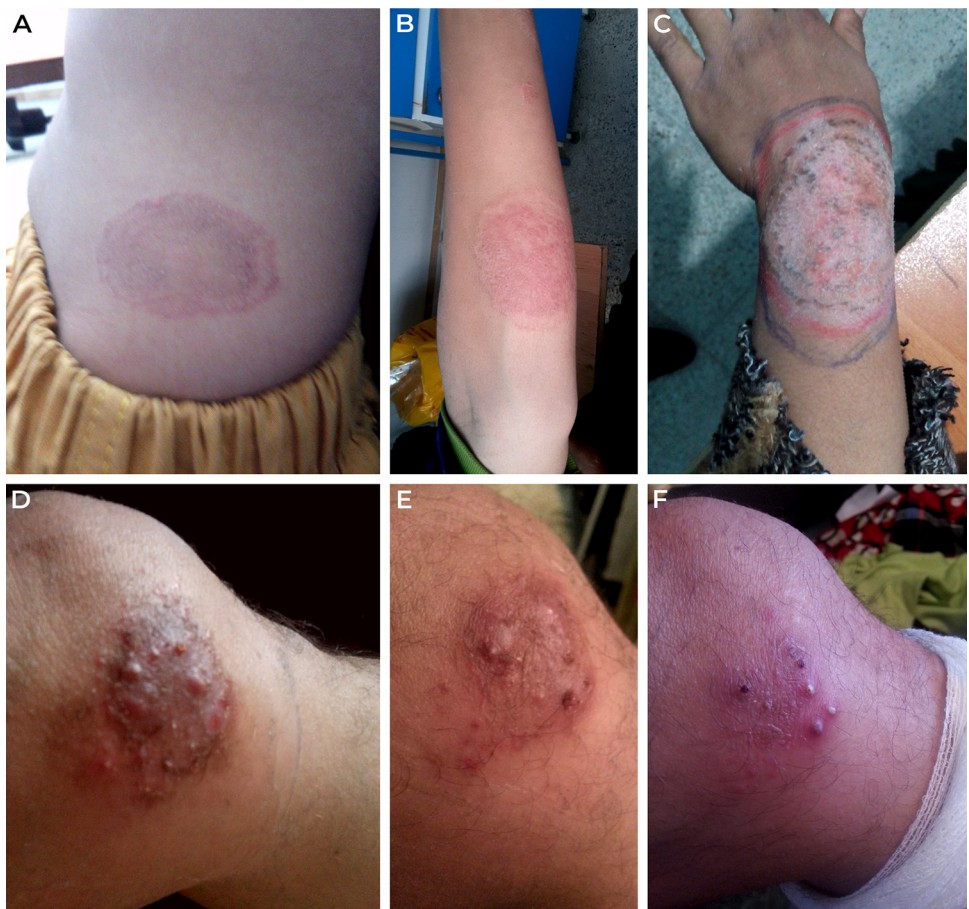

**FIG 2** Clinical manifestations of infections caused by *Trichophyton persicum* (A to C) and *Trichophyton spiraliforme* (D to F). Annular lesion with erythematous raised margins on the abdomen of a 2-year-old boy (isolate Ahv-525) (A); tinea corporis of the extensor surface of the right forearm with inflammatory erythema and scaling in a 16-year-old female (isolate Ahv-523) (B); an extended annular, scaly plaque on the extensoral aspect of the left forearm and wrist in a 31-year-old woman (isolate S366)—the highlighted regions are accentuated borders that had been manipulated by the patient (C); red-brown indurated plaque with several pustules, papules and crusts on the inner aspect of the left knee in a 43-year-old man (D); the same lesion after 1 week of terbinafine treatment (E); detail of pustules in the same patient (F).

A dermatological examination demonstrated a well-defined 6- by 3.5-cm erythematous pustular plaque with adherent yellowish crusts (Fig. 2D to F). A swab sample was taken from the pustules and crusts for mycological examination. Fungal mycelium was demonstrated on potassium hydroxide examination, and the agent was preliminarily identified based on morphological features as the zoophilic strain of *Trichophyton interdigitale* according to the concept of Heidemann et al. (32). The identification was subsequently corrected to *Trichophyton* sp. after ITS rDNA region sequencing.

Nobody else in the patient's surroundings had similar symptoms. Oral treatment with terbinafine (250 mg daily) in combination with clotrimazole (1% cream, two times a day) was initiated based on the result of the mycological examination. The pustules disappeared in 2 weeks after starting the treatment. A cure was achieved 4 weeks after the onset of the treatment, with only slight erythema remaining. No recurrence has been observed to date.

Later the patient mentioned that he owned a dog (German shorthaired pointer) that had been treated with antimycotics at a veterinary surgeon 2 months ago due to a lesion on its front paw above the ankle. Although the transmission of the infection from the dog cannot be confirmed due to the absence of an animal isolate, it is highly suspected.

**Phylogeny.** For the phylogenetic analysis, we used a previously published alignment consisting of 340 combined ITS, *gapdh*, *tubb*, and *tef1-α* sequences from members of the *T. benhamiae* complex (10); only the sequences of the presumed hybrid IHEM 25139 were

excluded. The alignment was enriched by available ITS and *tef1-α* sequences from *T. persicum*. Four genetic loci were only amplified in the viable ex-type isolates of both *T. persicum* and *T. spiraliforme*. In addition, one Iranian strain of *Trichophyton eriotrephon* was also added. The final alignment included 378 isolates and 2,374 characters, with 243 variable and 153 parsimony informative sites. The alignments, GenBank accession numbers, and origins of all 378 isolates are available in the Dryad Digital Repository at https://doi.org/10.5061/dryad .59zw3r275.

The phylogenetic analysis resolved both new species within the *T. benhamiae* clade sensu Čmoková et al. (10) of the *T. benhamiae* complex (Fig. 3). *Trichophyton spiraliforme* was placed in the sister position to *T. persicum* with a Bayesian posterior probability of 0.96, while the maximum-likelihood (ML) bootstrap was low (<70%). The relationships of these species with other taxa in the *T. benhamiae* clade are poorly resolved due to insufficient statistical supports, but the closest species are represented by *T. japonicum* and *T. europaeum*. The phylogenies based on the ITS rDNA region (all isolates) and 4 loci (only isolates having full sequence data available) are shown for comparison in Fig. S1 and S2 in the supplemental material. The topologies of these phylogenetic trees are similar, with some differences in deep nodes that gained low statistical support in all phylogenies.

*Trichophyton persicum* and *Trichophyton spiraliforme* can be differentiated from each other by their ITS rDNA (2 substitutions in the ITS2 region), *gapdh* (12 substitutions), and *tef1-α* gene sequences (1 substitution and 1 indel). The ITS rDNA and *gapdh* loci can also differentiate both species from the other members of the *T. benhamiae* complex. The sequence of the *tef1-α* gene of *T. spiraliforme* is identical to those of *T. europaeum* and *T. japonicum*, and it shows a single substitution and indel compared to that of *T. persicum*. The two new species share identical *tubb* sequences that differ by two substitutions from those of all other members of the *T. benhamiae* clade. The *tubb* locus shows overall low discriminatory power in the *T. benhamiae* complex, and all remaining members of the *T. benhamiae* clade have identical *tubb* genotypes, as demonstrated previously (10).

Some level of intraspecific variability was detected in the available ITS ($n = 37$) and *tef1-α* ($n = 8$) sequences of *T. persicum*; specifically, there were three substitutions and one indel in the ITS region and a single substitution in *tef1-α* gene sequences. The remaining two loci were only available for the ex-type strains, and consequently, the level of intraspecific variability, along with the precise numbers of species-specific substitutions/indels useful for their differentiation, needs to be confirmed in future studies when more strains will be available. Given the importance of the ITS rDNA region for routine dermatophyte identification, we summarized all species-specific substitutions and indels between all species pairs from the *T. benhamiae* clade in Table S1.

The genotype network of the *T. benhamiae* complex based on a combined alignment (only strains with all four loci available) is shown in Fig. 4, where hash marks on the connecting lines indicate the numbers of substitutions between species. Fig. 4 also shows the distribution of *MAT1-1-1* and *MAT1-2-1* gene idiomorphs across the *T. benhamiae* complex. In general, both idiomorphs are unequally distributed in most species of the complex, and one idiomorph is frequently absent, suggesting clonal spread of the majority of species. Data from more strains are needed for *MAT* idiomorph distribution assessment in the two newly described species.

**Microsatellite analysis.** The microsatellite analysis (Fig. 5) resolved both new species into proximity to *T. benhamiae* var. *benhamiae* cluster 2 and *T. benhamiae* var. *luteum*. Both *T. persicum* and *T. spiraliforme* formed independent distant lineages. The analysis separated different subpopulations of *T. benhamiae* var. *benhamiae*. Cluster 2 of *T. benhamiae* var. *benhamiae* sensu Čmoková et al. (10) consisted of two lineages related to *T. benhamiae* var. *luteum*, while cluster 1 grouped with *T. concentricum*. Consequently, *T. benhamiae* did not form a monophyletic group. This population genetic variability in *T. benhamiae* is, however, not reflected by standard sequence markers and multigene phylogeny (Fig. 3).

**Phenotype analysis.** The growth rate of *T. spiraliforme* is similar to the growth rates of other zoophilic species from the *T. benhamiae* clade with the so-called "white phenotype" (*T. europaeum*, *T. japonicum*, and *T. benhamiae* var. *benhamiae*). *Trichophyton spiraliforme*

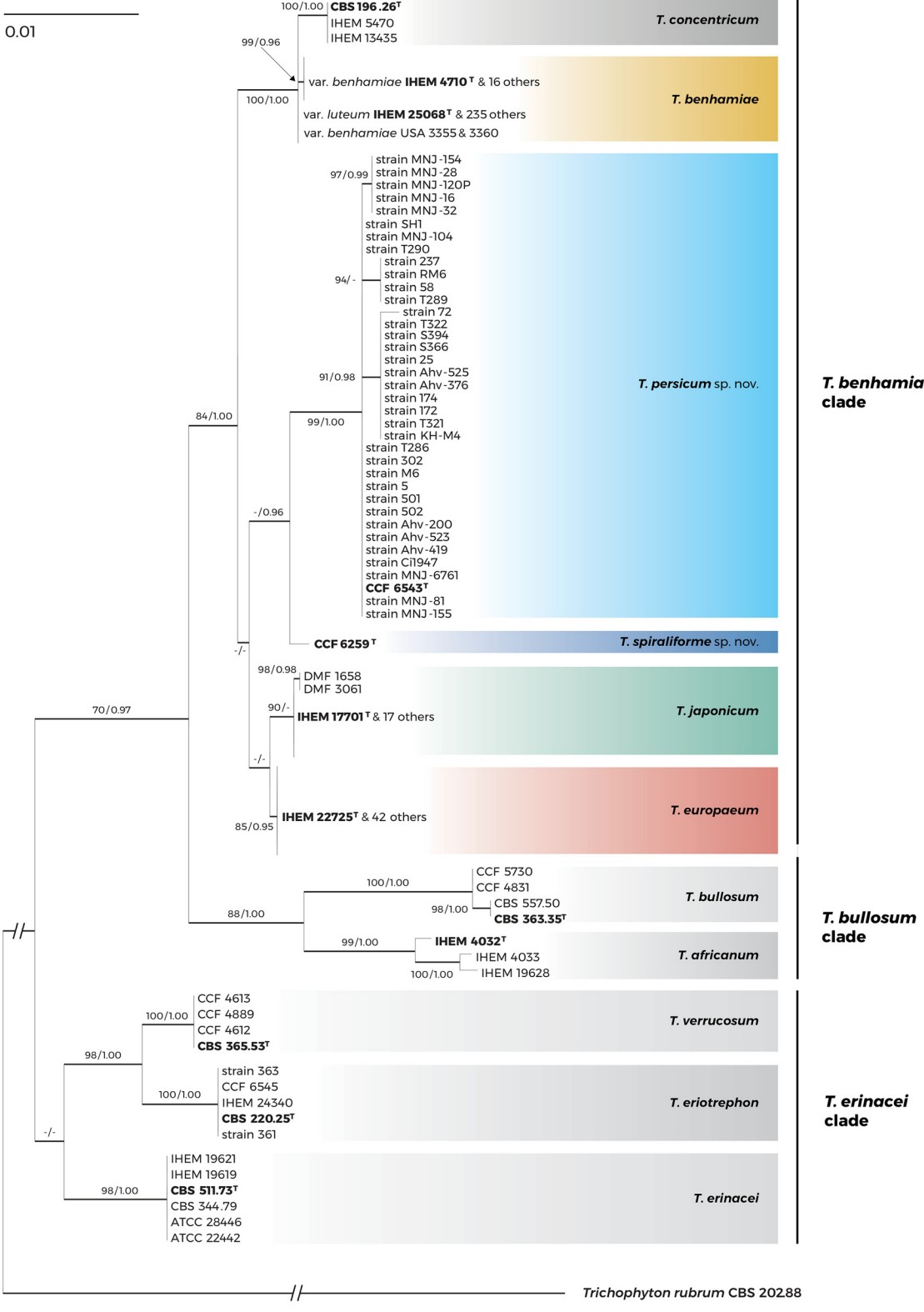

**FIG 3** A best-scoring maximum-likelihood tree (*gapdh*, *tubb*, ITS rDNA, and *tef1-α*) showing relationships of *Trichophyton spiraliforme* sp. nov. and *Trichophyton persicum* sp. nov. to other dermatophytes belonging to the *T. benhamiae* complex. Only support values exceeding bootstrap values of 70% and Bayesian posterior probabilities of 0.90, respectively, are shown. Ex-type isolates are designated by a superscript T. *Trichophyton rubrum* CBS 202.88 was used as the outgroup. The phylogenies based on the ITS rDNA region (all isolates) and 4 loci (only isolates having full sequence data available) are shown in the supplemental material (Fig. S1 and S2) for comparison.

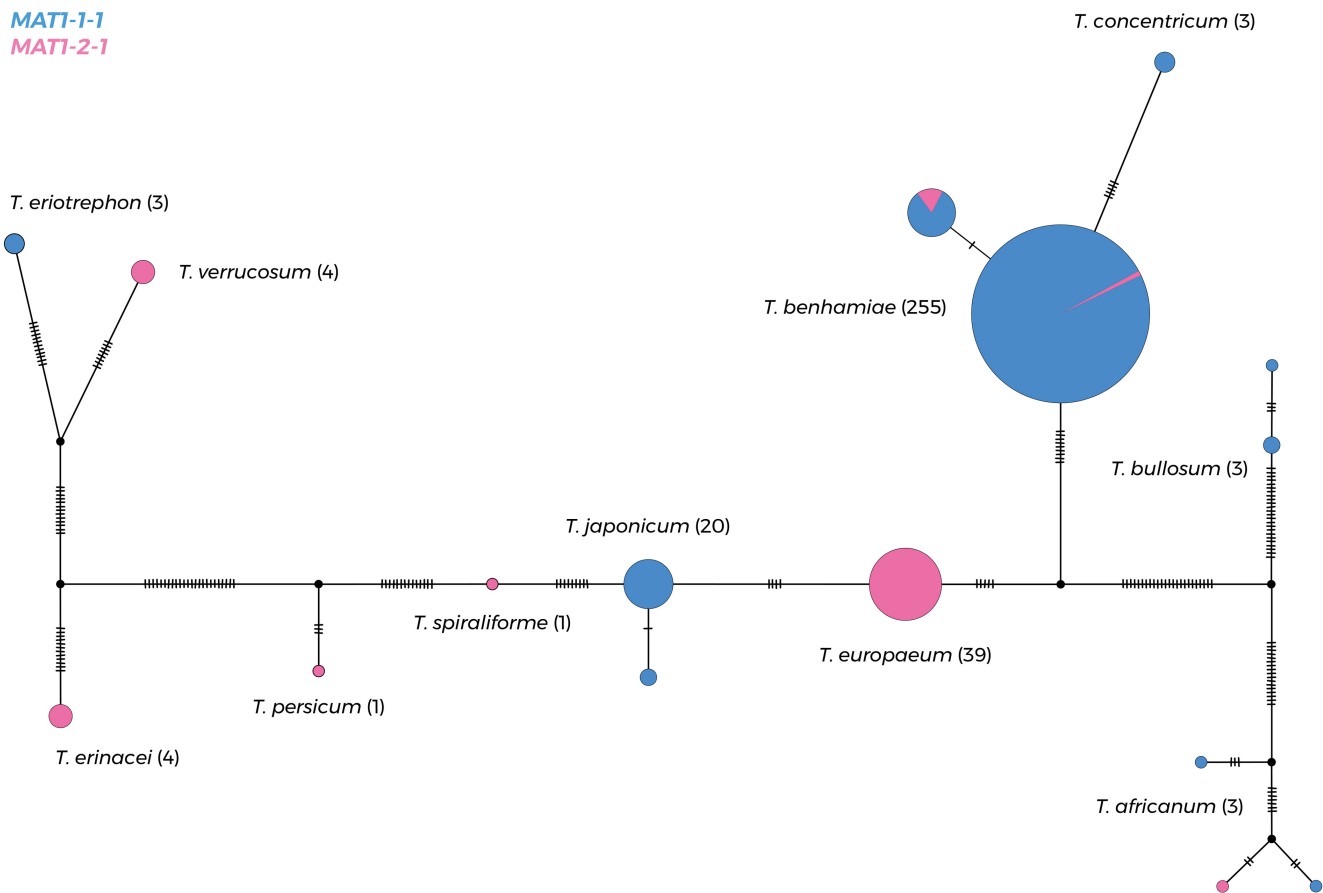

## MAT idiomorph

*MAT1-1-1*
*MAT1-2-1*

*T. concentricum* (3)

*T. eriotrephon* (3)

*T. verrucosum* (4)

*T. benhamiae* (255)

*T. bullosum* (3)

*T. japonicum* (20)

*T. spiraliforme* (1)

*T. europaeum* (39)

*T. persicum* (1)

*T. erinacei* (4)

*T. africanum* (3)

**FIG 4** Haplotype network based on multilocus data (ITS rDNA, *gapdh*, *tef1-α*, and *tubb* loci) showing the distribution of mating-type (*MAT*) gene idiomorphs in the *Trichophyton benhamiae* complex. Haplotypes are indicated by circles whose sizes correspond to the numbers of strains analyzed (data from Čmoková et al. [10], updated by new isolates from this study), with the numbers of strains given in parentheses after the species name, and hash marks on the connecting lines indicate substitutions (indels are excluded).

differs mainly by its restricted growth on potato dextrose agar (PDA). Slightly lower growth rates on PDA were also observed in some strains of *T. europaeum*, but not to such an extent. *Trichophyton persicum* grows restrictedly also but on all media tested, as observed in the ex-type strain. Limited data are available for several strains that were no longer viable and could not be used for detailed analyses. These strains (MNJ-81, S394, and KH-M4) were all characterized by slow growth on Sabouraud dextrose agar (SDA) with chloramphenicol and cycloheximide. Very restricted growth on SDA at 37°C also differentiates *T. persicum* from other closely related zoophilic species.

Both new species produce small microconidia whose dimensions do not deviate significantly from those of other *T. benhamiae* clade species (Fig. 6). Microconidia were abundantly produced and predominantly clavate in *T. spiraliforme*, while relatively poor production was observed in *T. persicum* and the conidia were frequently irregular and diverse in shape (Fig. 7). In this aspect, *T. persicum* resembles *T. eriotrephon*. *Trichophyton spiraliforme* produced a large number of spiral hyphae that were observed already in 7-day-old colonies and were very abundant after 21 days of cultivation. Although this is a very subjective character, such an extensive production of spiral hyphae is uncommon in *T. benhamiae* clade species and is rather typical for *T. mentagrophytes*.

## TAXONOMY

*Trichophyton persicum* Rezaei-Matehkolaei, Cmokova & Hubka sp. nov. MycoBank accession number MB839323; Fig. 7.

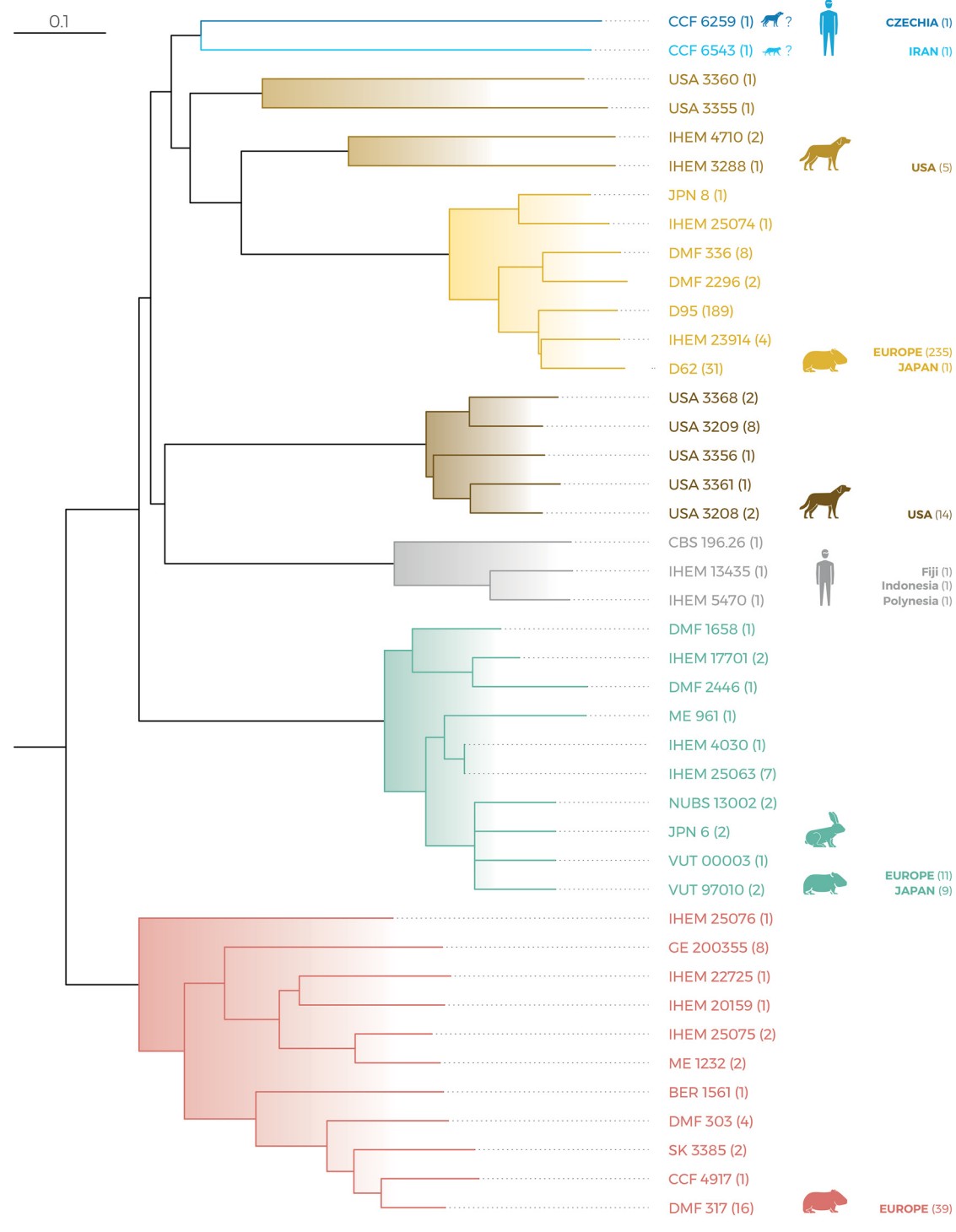

**FIG 5** The population structure of the *Trichophyton benhamiae* clade, based on 10 microsatellite loci and 319 isolates (data from Čmoková et al. [10], updated by new isolates from this study). The neighbor-joining tree was calculated from the multilocus microsatellite profiles using the Jaccard distance matrix measure in FAMD 1.3 (86). The assignment of strains to main clusters and species is indicated by different colors; clones were discarded from the analysis. The number of isolates representing each haplotype is indicated in parentheses following the isolate number. The labels of each cluster show the geographic origin of strains with the total number of isolates and the main primary host(s).

µm / length of microconidia

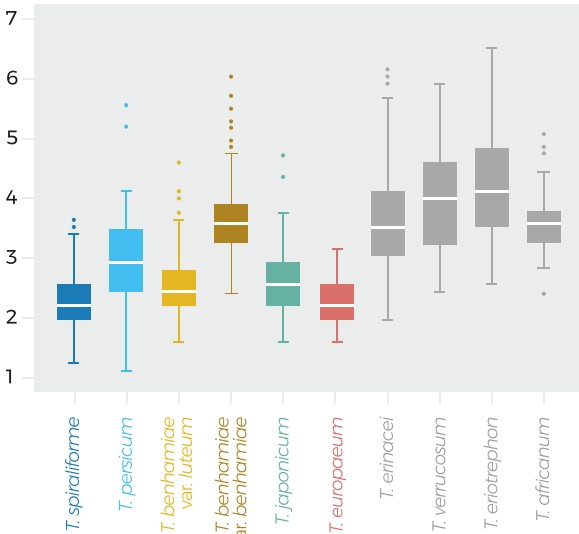

µm / width of microconidia

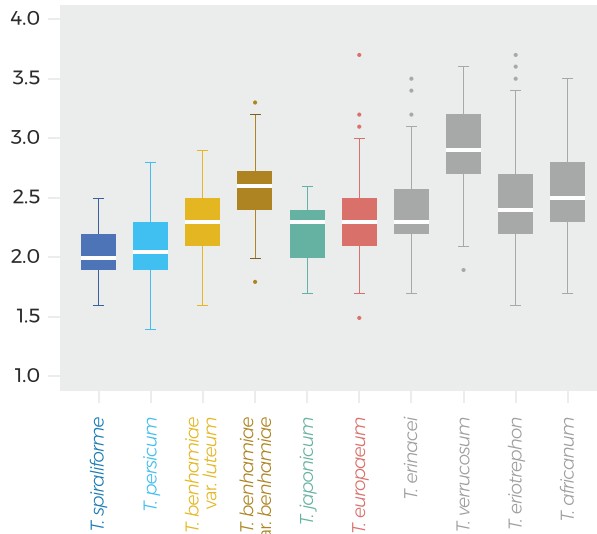

**FIG 6** Lengths and widths of microconidia in taxa belonging to the *Trichophyton benhamiae* complex, including two new species, *T. spiraliforme* and *T. persicum*. The horizontal lines indicate mean values and interquartile ranges, whiskers span the 5% and 95% percentiles, and circles represent extreme outliers.

Etymology: referring to the origin of the examined strains, Persia, the ancient name of Iran.

Holotype: PRM 954724, a dried herbarium specimen (isotypes PRM 954725 and PRM 954726); Iran, Qaen, tinea capitis (ectothrix), 5-year-old girl, 2017, M. J. Najafzadeh, ex-holotype living culture CCF 6543 = MJN-91.

Vegetative hyphae smooth, septate, hyaline, 1 to 2.5 µm in diameter (mean ± standard deviation [SD], 1.5 ± 0.3). Conidiophores poorly differentiated from vegetative hyphae and represented by conidiogenous hyphae, unbranched or with sparse lateral branches, usually densely septate. Microconidia are rare, borne sessile on hyphae or on short lateral protrusions of hyphae, irregularly shaped, usually clavate, ovoid, or pyriform, less commonly barrel shaped or irregular, occasionally released together with fragile lateral protrusions of hyphae, 2 to 6 (3.6 ± 0.7) by 1.5 to 3 (2 ± 0.3) µm. Macroconidia not observed. Chlamydospores present. Spiral hyphae not observed even after 21 days of cultivation. Sexual morph is unknown.

Colonies on malt extract agar (MEA; Oxoid, Basingstoke, UK) in 7 days at 25°C attained 9 to 11 mm in diameter (ø = 11 mm), white (#F5F5F0) to pale orange yellow (#F3E5AB), raised in the center, velvety to cottony, edge entire, reverse light orange yellow (#F8DE7E) to vivid orange yellow (#F6A600). Colonies on SDA in 7 days at 25°C attained 10 to 12 mm in diameter (ø = 11 mm), white (#F5F5F0), pale orange yellow (#FFF587), or light yellowish brown (#E3D6A1); dark brown diffuse pigment produced into medium after 3 to 4 weeks of cultivation. Colonies on PDA in 7 days at 25°C attained 8 to 10 mm in diameter (ø = 8 mm), white (#F5F5F0) to pale orange yellow (#F3E5AB), raised, velvety or cottony, edge entire, reverse light orange yellow (#F8DE7E). Colonies on SDA in 7 days at 30°C attained 15 to 18 mm in diameter (ø = 17 mm) and at 37°C attained 5 to 8 mm in diameter (ø = 6 mm).

*Trichophyton persicum* differs from zoophilic members of the *T. benhamiae* clade by its low growth rates on all media examined and restricted growth at 37°C. Other slow-growing species in this clade are the anthropophilic *T. concentricum* and zoophilic *T. benhamiae* var. *luteum*. The latter differs from *T. persicum* by a yellow colony reverse, the shape of the microconidia, and ecology. *Trichophyton persicum* most closely resembles *T. eriotrephon* by the variability in the shape of microconidia and restricted growth on 37°C. These species can be differentiated by conidium length, which is longer in *T. persicum* than in *T. eriotrephon*. The macromorphology of *T. persicum* may resemble that of *T. verrucosum* by slow growth and colony appearance, but the species differ in micromorphology, as *T. verrucosum* usually

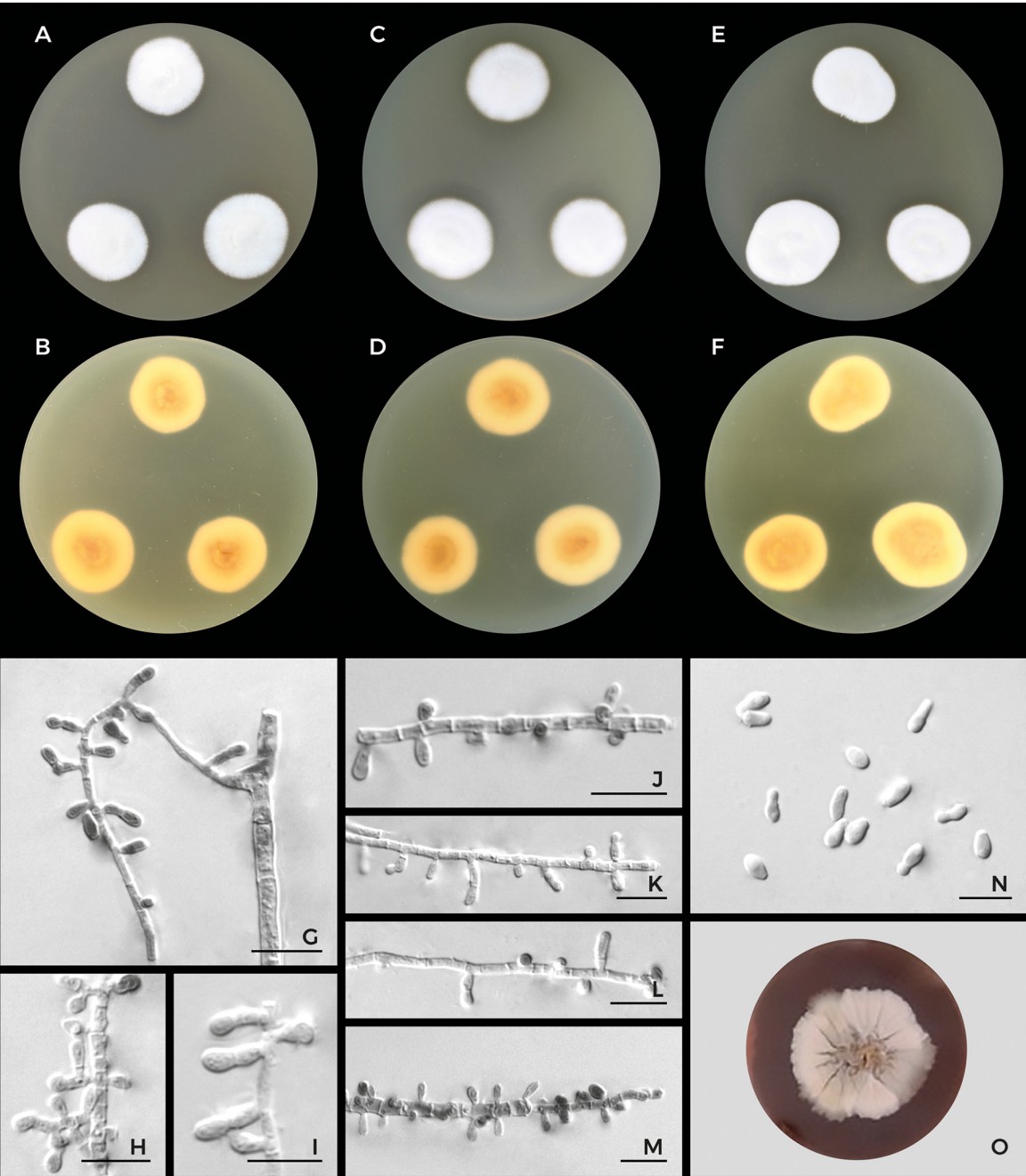

**FIG 7** Macromorphology and micromorphology of *Trichophyton persicum* CCF 6543. Colonies after 2 weeks of cultivation at 25°C on SDA (A) and the reverse (B), on MEA (C) and the reverse (D), and on PDA (E) and the reverse (F); conidiophores (G to M); microconidia and intercalary conidia with varying shapes (N); and a colony with brown diffuse pigment produced into the medium (strain MJN-16) after 2 weeks at 25°C on SDA supplemented with cycloheximide and chloramphenicol (O). Scale bars, 5 μm.

produces only chlamydospores and no microconidia. The only viable isolate examined by us exhibited a *MAT1-2-1* mating type gene idiomorph.

*Trichophyton spiraliforme* Cmokova, Kuklova & Hubka, sp. nov. MycoBank accession number MB839324; Fig. 8.

Etymology: referring to the abundant production of spiral hyphae.

Holotype: PRM 954616, a dried herbarium specimen (isotype PRM 954617); Czech Republic, Hostouň near Prague, knee skin (tinea corporis profunda), 43-year-old man, 2017, J. Stará & I. Kuklová, ex-holotype living culture CCF 6259 = SK 4179/17.

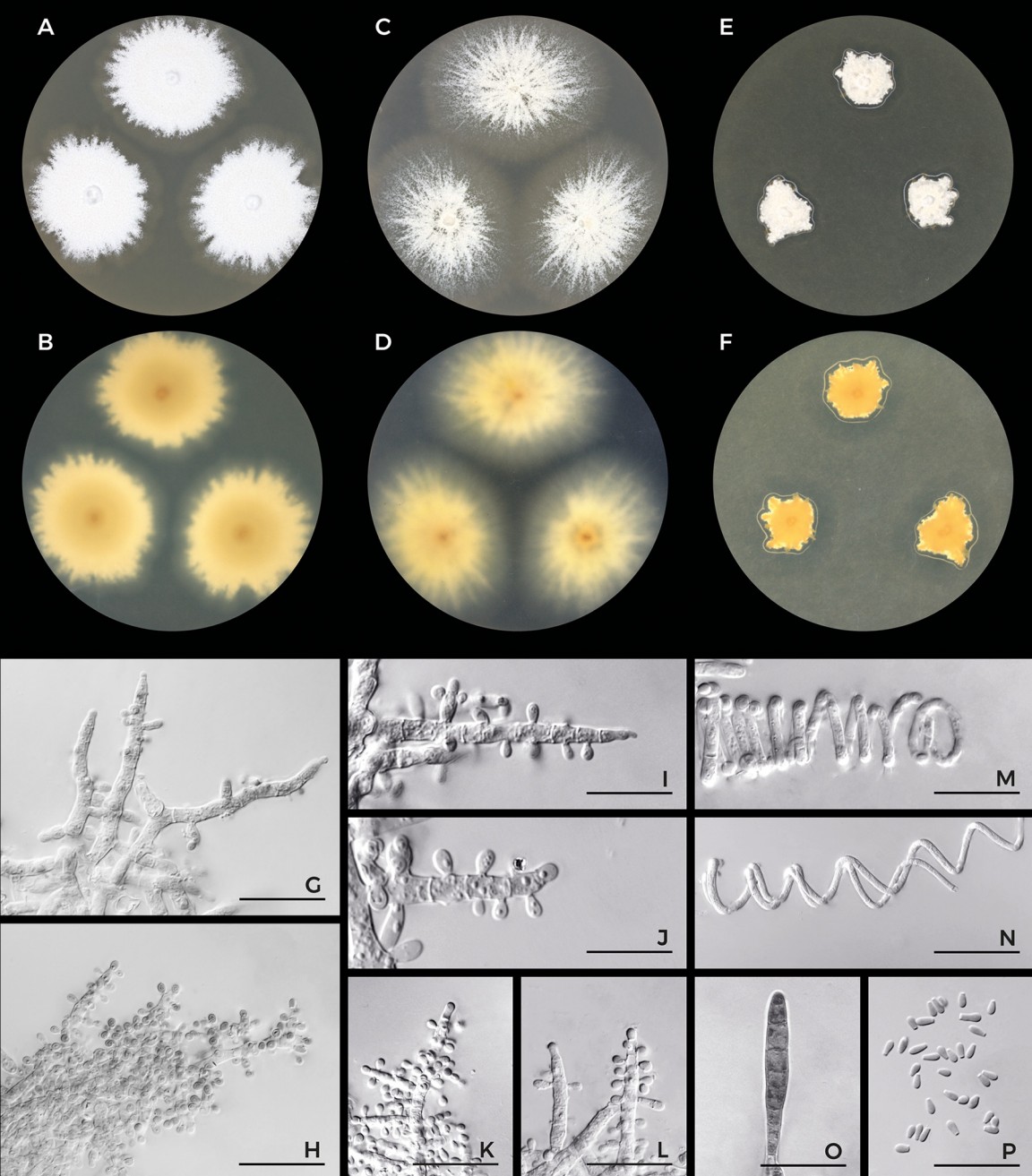

**FIG 8** Macromorphology and micromorphology of *Trichophyton spiraliforme* CCF 6259. Colonies after 2 weeks of cultivation at 25°C on SDA (A) and the reverse (B), on MEA (C) and the reverse (D), and on PDA (E) and the reverse (F); conidiophores (G to L); spiral hyphae (M and N); macroconidium (O); and microconidia (P). Scale bars, 10 µm.

Vegetative hyphae smooth, septate, hyaline, 1 to 3.5 µm in diameter (mean ± SD, 2.2 ± 0.7). Conidiophores both poorly differentiated from vegetative hyphae and represented by conidiogenous hyphae with sparse to numerous short lateral branches, occasionally well differentiated and branched in a pyramidal (grape-like) pattern. Microconidia abundant, pyriform to clavate, 2.5 to 4 (3 ± 0.4) × 2 to 2.5 (2 ± 0.2) µm. Macroconidia rare, borne laterally or terminally on hyphae, consisting of 5 to 8 cells (median = 5), 25.5 to 33.5 (26.8 ± 3.5) by 4.3 to 6 (5 ± 0.6) µm, elongated, cigar-shaped, clavate, with a tapering rounded apex and truncate base. Chlamydospores present. Spiral hyphae observed after 7 days of cultivation, abundantly present after 21 days. Sexual morph is unknown.

Colonies on MEA in 7 days at 25°C attained 15 to 35 mm in diameter (ø = 21 mm), granular, white (#F2F3F4) to light yellow (#FAD6A5), flat, edge irregular, submerged, reverse pale orange yellow (#FAD6A5) to brilliant orange yellow (#FFC14F). Colonies on SDA in 7 days at 25°C attained 14 to 17 mm in diameter (ø = 15 mm), white (#F2F3F4), powdery, umbonate, edge irregular, submerged, reverse deep brown (#593319) in the center, light orange yellow (#FBC97F) in the marginal parts. Colonies on PDA in 7 days at 25°C attained 8 mm in diameter (ø = 8 mm), white (#F2F3F4) to light yellow (#FAD6A5), granular, occasionally with filamentous sectors, raised, reverse pale orange yellow (#FAD6A5) to brilliant orange yellow (#FFC14F). Colonies on SDA in 7 days at 30°C attained 25 to 27 mm in diameter (ø = 26 mm) and at 37°C attained 20 to 24 mm in diameter (ø = 22 mm).

The morphology of *T. spiraliforme* most closely resembles those of related species from the *T. benhamiae* complex (*T. europaeum*, *T. japonicum*, and *T. benhamiae* var. *benhamiae*) and *T. mentagrophytes* by colony texture and numerous relatively small microconidia borne on conidiophores branched in a grape-like pattern. It differs from the species mentioned by its restricted growth on PDA. The only isolate examined by us exhibited a *MAT1-2-1* mating type gene idiomorph.

## DISCUSSION

In our previous research, we used a polyphasic approach based on morphology, physiology, ecology, clinical picture, and molecular data, including noncoding regions, to resolve the taxonomy of the *T. benhamiae* complex (10, 33). This approach supported the inclusion of nine taxa in the complex, including three new species and one new variety. A similar approach was applied to resolve the positions of tentative new species in this study and led to the description of *T. persicum* and *T. spiraliforme*. While the DNA sequence data assigned both species to the vicinity of *T. europaeum* and *T. japonicum* (Fig. 3), microsatellite data rather showed their relatedness with North American strains of *T. benhamiae* var. *benhamiae* cluster 2 (Fig. 5), mostly isolated from dogs. Based on microsatellite data, North American strains of *T. benhamiae* var. *benhamiae* do not form a monophyletic group due to high variability, which is not reflected in standard DNA sequence markers. Thorough sampling and obtaining a larger number of North American *T. benhamiae* strains, as well as *T. spiraliforme* and *T. persicum* strains, would probably better elucidate the population structure of the whole species complex. Because only one strain each of *T. spiraliforme* and *T. persicum* was available for microsatellite analysis, we cannot draw any conclusions about the population structure of these species. However, they formed distant and independent lineages separated from each other and the population of *T. benhamiae*.

Reliable species identification of these new species can be achieved by sequencing of ITS rDNA (Table S1) and *gapdh* loci, as they contain species-specific substitutions. Identification can also be complemented by routine morphological examination, because the new species show unique phenotypic features. Namely, both species have specific growth parameters: *T. spiraliforme* grows slowly on PDA, while *T. persicum* grows slowly on all media examined at 25°C and grows poorly at 37°C. In addition, *T. persicum* has the characteristic shape of microconidia and sporulates poorly, which is rather atypical for zoonotic pathogens in *T. benhamiae* complex. Other features, such as extensive and early production of spiral hyphae in *T. spiraliforme* or production of pigments into medium, are rather subjective and should be verified on a larger number of strains in the future. Selected characteristics useful for differentiation of particular species in the *T. benhamiae* complex are shown schematically in Fig. 9.

The rise of antifungal resistance in some dermatophytes, especially terbinafine resistance in members of the *T. mentagrophytes* and *Trichophyton rubrum* complexes, is a significant clinical problem nowadays, since it can lead to treatment failure (34–36). No antifungal resistance has been reported in members of the *T. benhamiae* complex (37). Several *T. persicum* strains have been tested in previous studies under the name *T. benhamiae* (38, 39) and reidentified in this study. The results showed that the antifungal susceptibility pattern of *T. persicum* does not differ from those of other species of the

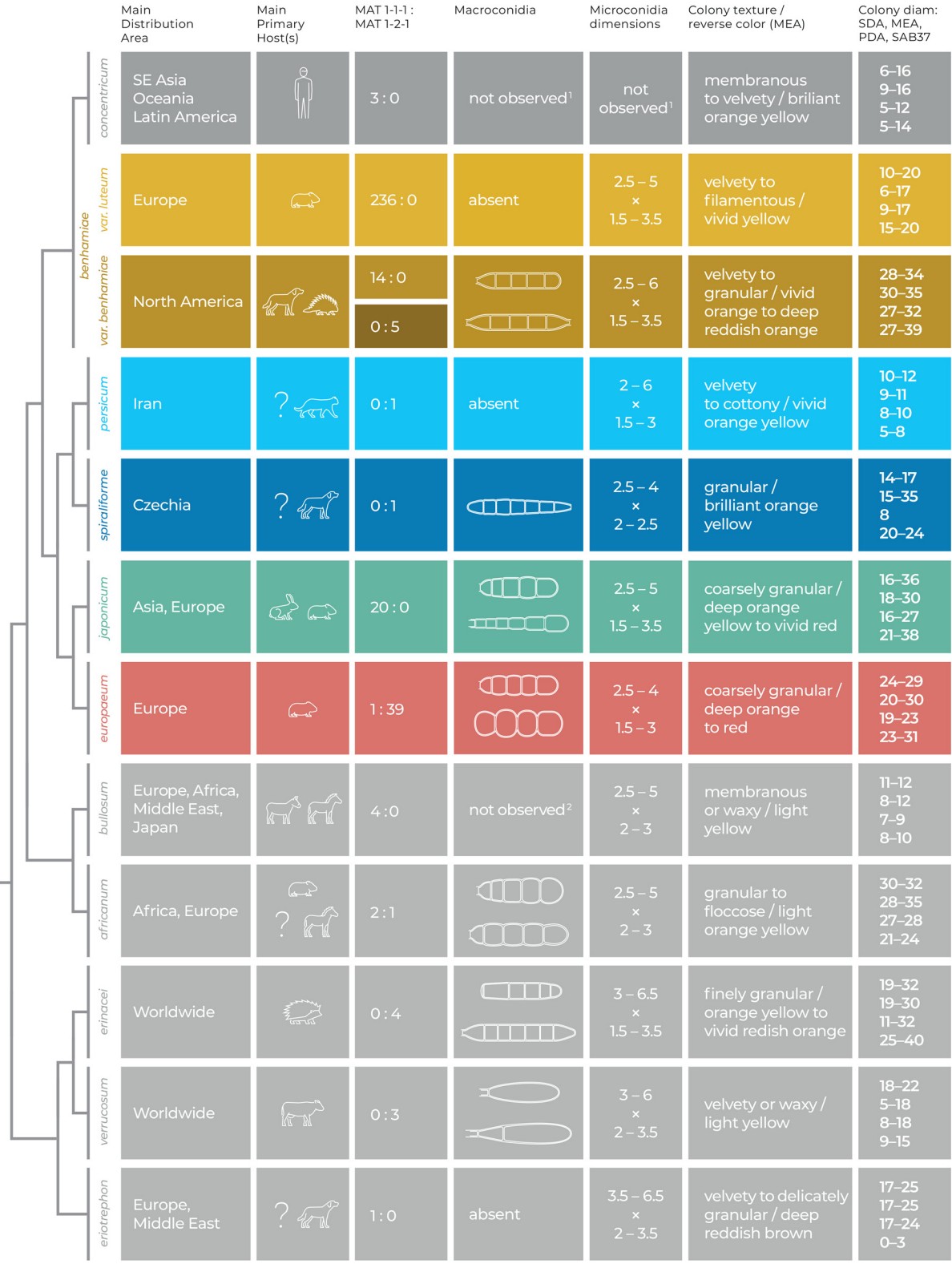

**FIG 9** Overview of selected phenotypic features and ecological data plotted on the simplified phylogeny of the *Trichophyton benhamiae* species complex. The main primary host(s) of particular species are shown as icons; uncommon occasional hosts are omitted; a question mark means that the host spectrum is little known and is based on only one or few isolations from animals/humans. Explanation of superscript numbers: 1, macroconidia were observed in more than one study (12, 88); 2, macroconidia were observed by Lebasque (89) under specific conditions.

*T. benhamiae* clade. Luliconazole, efinaconazole, and terbinafine were the most potent antifungals against species from the *T. benhamiae* complex (37). Successful treatment of tinea corporis due to *T. spiraliforme* by terbinafine in this study also indicates susceptibility to this commonly used antifungal agent.

The epidemiology of dermatophytosis in Iran shows local specifics in comparison with its epidemiology in European or Asian countries (40). Compared to European countries, where onychomycosis and tinea pedis are the most common clinical entities and older age groups usually predominate among patients (1), typical Iranian patients belong to younger age groups, and tinea corporis (including tinea cruris) and then scalp infections are the most prevalent types (39, 41, 42). Onychomycosis and tinea pedis contribute to lesser extents to the overall spectrum of dermatophytosis in Iran, but they start to prevail in the Northern, Central, and Western regions (43–47). Aside from rare cases due to *T. verrucosum* and *T. eriotrephon*, zoophilic species from the *T. benhamiae* clade (*T. benhamiae*, *T. europaeum*, *T. japonicum*, and *T. persicum*) are marginal infectious agents that contribute to the spectrum of all dermatomycosis agents only at rates of 0.5 to 5%. Based on long-term epidemiological data and the sequence data available in the GenBank database, *T. persicum* seems to be the dominant species from the clade in Iran, responsible for almost all cases and mostly restricted to a relatively small area in South, Southwestern, and Eastern Iran (Fig. 1). But it must be added that no extensive epidemiological studies have been performed in neighboring countries. In terms of case frequencies, *T. persicum* is followed by *T. eriotrephon* and then by extremely rare cases due to *T. benhamiae*, *T. europaeum*, and *T. japonicum* (39, 42, 48, 49) which were retrospectively identified based on the data deposited in GenBank (10).

The epidemiological situation in Europe in terms of the occurrence of *T. benhamiae* complex pathogens is in striking contrast to that in Iran. *Trichophyton benhamiae* var. *luteum* belongs to the most important zoonotic agents of dermatophytosis (19, 20, 50, 51), and cases caused by other species, such as *T. europaeum* and *T. japonicum*, are also relatively common (10). These differences are probably caused by local socioeconomic and cultural practices and different spectra of farmed and pet animals that influence the prevalences of particular species. Although many dermatophytes have global distribution, there are some species restricted to specific geographical areas or with poorly known distribution. Such a phenomenon is well documented, for instance, in species from the *T. rubrum* complex, *Microsporum ferrugineum*, and *Microsporum audouinii* (40, 52–54). There are also several examples in the *T. benhamiae* complex; namely, *T. benhamiae* var. *luteum*, *T. europaeum*, and *T. japonicum* have their main distribution areas in Eurasia, whereas *T. benhamiae* var. *benhamiae* seems to be mostly restricted to North America and *T. concentricum* is typical for tropical regions of Oceania, Southeast Asia, and Central and South America (10). However, international travel and animal transport erase to a large extent the original geographic areas of distribution. *Trichophyton spiraliforme* is apparently a rare species in the human clinical material, at least in Europe, where we have large amounts of molecular data available from multiple surveys (17, 19, 50, 55). On the other hand, it is not known how frequent it can be in veterinary samples where we only have limited data.

It is assumed that both new species are zoophilic pathogens based on their phylogenetic relationships with other zoophilic species and clinical manifestations in human patients. Both pathogens were predominantly isolated from cases of human tinea corporis. In the case of *T. spiraliforme*, a dog is suspected to be the source of dermatophytosis in the patient examined, while in *T. persicum*, there is evidence in one case of isolation from a stray cat.

In general, the spectrum of dermatophytes in dogs is poorly known in Czechia. Data available from other European countries show that canine dermatophytosis is mainly caused by *Microsporum canis*, *T. mentagrophytes*, or *Nannizzia* spp. (56–60). In general, cases of canine dermatophytosis by *T. benhamiae* complex are rarely reported in Europe, and the majority of cases were described in North America and caused by *T. benhamiae* var. *benhamiae* (10, 61). This variety is related to *T. spiraliforme* based on microsatellite data, which further supports the hypothesis about transmission from a dog.

*Microsporum canis* is the main cause of feline dermatophytosis worldwide, and some areas of Iran are not an exception. This pathogen is responsible for up to 95% of feline infections in some studies, and has a prevalence of 26% in asymptomatic cats in Iran. The remaining cases are usually attributed to *T. mentagrophytes*, *T. verrucosum*, and *N. gypsea* (62–66). However, *T. verrucosum* has been reported in other studies as the main cause, with prevalences of up to 14.5% in asymptomatic cats (67). Feline dermatophytosis due to *T. verrucosum* is very unusual, and there are just a few cases reported from other countries besides Iran (68). Although it is known that this species, typically associated with cattle and other ruminants, can sporadically infect any mammal and even birds (22), it could be expected that the course of infections should be rather serious than asymptomatic in cats. There could be several explanations. The isolation from an asymptomatic cat may indicate fomite carriage from exposure to a contaminated environment. Alternatively, we may assume that some cases of feline dermatophytosis could be caused by the slow-growing and superficially somewhat similar *T. persicum* and not by *T. verrucosum*, because almost all feline isolates in previous studies from Iran were solely identified by morphological characteristics. The only *T. persicum* isolate from cats examined by us originates from the study of Moosavi et al. (67), who recovered 15 dermatophyte isolates from the skin and hair of 103 asymptomatic stray cats in Meshkin-shahr (Northwestern Iran). Among these strains, 13 were designated as *T. verrucosum* and 2 as *T. mentagrophytes* based on macro- and micromorphological characters. One *T. verrucosum* isolate was subsequently subjected to ITS rDNA sequencing and has now been found to be identical with *T. persicum* and not *T. verrucosum*.

**Conclusion.** The increasing number of epidemiological studies substantiated by species identification using molecular data and the fast-growing amount of sequence data in the global databases bring valuable information about ecology, host spectra, and distribution of dermatophyte species that can also lay the groundwork for taxonomic studies. In this study, we were able to retrospectively identify and describe two new *Trichophyton* species based on long-term epidemiological studies on dermatophytosis that were carried out in Iran and Czechia. These presumably zoophilic members of the *T. benhamiae* complex were proposed based on a polyphasic approach, and diagnostic options for their identification are presented based on both molecular and morphological characters. Further research is needed to refine our knowledge about the host spectrum and geographical distribution of these species.

## MATERIALS AND METHODS

**Strain studies.** We conducted retrospective identification of Czech and Iranian isolates belonging to the *T. benhamiae* complex according to the recent taxonomy (10). All strains were isolated during epidemiological surveys investigating the spectrum of dermatophytosis in different areas in Iran and Czechia during the years 2008 to 2020. Dermatophytosis agents were isolated from skin, nails, nail debris, and scalp samples and identified using molecular methods. In general, ITS rDNA barcode sequences were used for identification, and in addition, a partial *tef1-α* gene, encoding translation elongation factor 1-α, was amplified in some Iranian strains. Detailed methodologies were described in the particular studies from which the isolates originated: epidemiological surveys were conducted in Tehran during 2008 to 2010 and 2012 to 2013 (44, 45), Shiraz during 2017 to 2019 (39), Southern Iran during 2012 to 2014 (42), Mashhad (North-Eastern Iran) during 2014 to 2015 (48), Isfahan during 2011 to 2012 (69), Khuzestan province during 2013 to 2014 (49), Qaen in 2017 (unpublished), and the various regions in Czechia during 2011 to 2020 (19, 70).

Ex-type isolates of the new species were deposited into the Culture Collection of Fungi (CCF), Department of Botany, Charles University, Prague, Czech Republic, and dried herbarium specimens (holotypes and isotypes) were deposited into the herbarium of the Mycological Department, National Museum in Prague, Czech Republic (PRM).

**Molecular studies.** In this study, the molecular studies were restricted to viable isolates, which were represented by ex-type strains of the newly described species. The genomic DNA was extracted from 7-day-old colonies using a fungal/bacterial miniprep kit (Zymo Research, Irvine, CA, USA). The quality was evaluated by using a NanoDrop 1000 spectrophotometer. The ITS rDNA region (ITS1–5.8S–ITS2 cluster) was amplified using the primer set ITS1F and ITS4 (71, 72), a partial *gapdh* gene, encoding glyceraldehyde-3-phosphate dehydrogenase, was amplified with primers GPDF and GPDR (73), a partial *tubb* gene, encoding β-tubulin, with primers Bt2a and Bt2b (74), and a partial *tef1-α* gene with primers EF-DermF and EF-DermR (75). The primer pair MF3 and MF6 was used for the detection of the *MAT1-1-1* idiomorph of the mating type gene (*MAT*), encoding a protein with an alpha domain motif, and primers TmHMG3S and TmHMG3R were used for the detection of the *MAT1-2-1* idiomorph, encoding a regulatory protein

with an HMG (high-mobility group) DNA-binding motif (76, 77). The reaction mixture volume of 20 $\mu$l contained 1 $\mu$l (50 ng ml$^{-1}$) of DNA, 0.3 $\mu$l of both primers (25 pM ml$^{-1}$), 0.2 $\mu$l of MyTaq polymerase, and 4 $\mu$l of 5× MyTaq PCR buffer (Bioline, London, UK). The PCR conditions followed the protocol described by Hubka et al. (78). PCR product purification followed the protocol of Réblová et al. (79). Automated sequencing was performed at Macrogen Sequencing Service (Amsterdam, The Netherlands) using both terminal primers.

**Phylogeny.** Alignments of the ITS, *gapdh*, *tubb*, and *tef1-α* regions were performed using the FFT-NS-i option implemented in MAFFT online (80). The alignments were trimmed and concatenated and then analyzed using maximum-likelihood (ML) and Bayesian inference (BI) methods. Suitable partitioning schemes and substitution models (Bayesian information criterion) were selected using a greedy strategy implemented in PartitionFinder 2 (81), with settings allowing introns, exons, codon positions, and segments of the ITS region to be independent data sets. The partitioning scheme (4 partitions) and substitution models for the ML analysis were as follows: the TrNef+G model was proposed for ITS1 and ITS2; the JC model for the 5.8S region, 1st-codon positions of *tubb*, and 2nd-codon positions of *gapdh*, *tubb*, and *tef1-α*; the F81+I model for the 1st-codon positions of *gapdh* and *tef1-α*; and the HKY+I model for the 3rd-codon positions of *gapdh*, *tef1-α*, and *tubb*. The ML trees were constructed with IQ-TREE version 1.4.4 (82), with nodal support determined by nonparametric bootstrapping with 1,000 replicates. Bayesian posterior probabilities were calculated using MrBayes 3.2.6 (83). The optimal partitioning scheme (4 partitions) and substitution models were identical to the ML analysis, with the exception of the first partition, for which the K80+G model was selected. The analysis ran for 10$^7$ generations, two parallel runs with four chains each were used, every 1,000th tree was retained, and the first 25% of trees were discarded as burn-in. The convergence of the runs and effective sample sizes was checked in Tracer version 1.6. The trees were rooted with *Trichophyton rubrum*.

**Haplotype network.** A haplotype network was constructed based on the combined ITS, *gapdh*, *tubb*, and *tef1-α* alignment; only isolates with full data available were retained. Variable position were extracted from the combined alignment and the TCS algorithm (84) implemented in the program PopART (85) was used to generate the haplotype network.

**Microsatellite analysis.** A multiplex panel consisting of 10 microsatellite markers developed for members of the *T. benhamiae* clade was used (10). The PCR mixture volume of 5 $\mu$l contained 50 ng of genomic DNA, 0.5 $\mu$l of a mixture of primers (final concentration 0.1 $\mu$M each primer), and 2.5 $\mu$l of multiplex PCR master mix (Qiagen, Hilden, Germany). The PCR conditions followed the manufacturer's instructions. The PCR products (diluted in water in a 1:20 ratio) were mixed with 10 $\mu$l of deionized formamide and 0.2 $\mu$l of the GeneScan 600 LIZ size standard and denatured for 5 min at 95°C, followed by analysis on an ABI 3100 Avant genetic analyzer. A binary and allele data matrix was created using GeneMarker 1.51 software (SoftGenetics, LLC, State College, PA, USA) and used to estimate the similarities between individuals using Jaccard's similarity coefficient calculation in the program FAMD (86). A neighbor-joining tree based on Jaccard's similarity coefficient matrix was constructed using the same software.

**Phenotype.** Micromorphological characteristics were recorded at least 35 times for each feature and documented using an Olympus BX-51 microscope. The macromorphology of the colonies was documented on MEA (Oxoid, Basingstoke, UK), potato dextrose agar (PDA; Himedia, Mumbai, India), and Sabouraud dextrose agar (SDA; Himedia, Mumbai, India) at 25, 30, and 37°C. The colonies were documented using a Canon EOS 500D camera. Color names were determined using the ISCC-NBS centroid color charts (87; hexadecimal color codes assigned independently according to the website are listed in the parentheses https://coolors.co).

**Data availability.** The DNA sequences obtained in this study were deposited into the GenBank database (www.ncbi.nlm.nih.gov) under accession numbers MW936599 to MW936628, MW959139 to MW959143, MZ314457, MZ320340, MZ320335, and MZ320330. The alignments are available in the Dryad Digital Repository at https://doi.org/10.5061/dryad.59zw3r275. The names of newly described species were deposited in MycoBank (MB839323 and MB839324).

## SUPPLEMENTAL MATERIAL

Supplemental material is available online only.
**SUPPLEMENTAL FILE 1**, PDF file, 0.2 MB.

## ACKNOWLEDGMENTS

The project was supported by the Czech Ministry of Health (grant number NU21-05-00681), a grant-in-aid for JSPS research fellow (grant no. 20F20772), Charles University Research Centre program no. 204069, and Czech Academy of Sciences long-term research development project (grant number RVO 61388971). M. J. Najafzadeh was supported by the elite researcher grant committee under award number 958797 from the National Institutes for Medical Research Development (NIMAD), Tehran, Iran. Vit Hubka is grateful for the support from the Japan Society for the Promotion of Science (postdoctoral fellowships for research in Japan—standard).

We are very grateful to Jan Karhan for the concept of data visualization and help with the graphical adjustments of analysis outputs. We thank Milada Chudíčkova and Soňa Kajzrová for their invaluable assistance in the laboratory, and Peter Mikula for

research support. We are also grateful to Simin Taghipour, Bahram Ahmadi, and Rasoul Mohammadi for their efforts in data collection.

The research reported in this publication was part of the long-term goals of the ISHAM working group Onygenales.

We declare no conflicts of interest.

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
