## [Reviewer comments · Microbiology Spectrum]

**Microbiology
Spectrum**

Discovery of new Trichophyton members, *T. persicum* and *T. spiraliforme* spp. nov., as a cause of highly inflammatory tinea cases in Iran and Czechia

Adéla Čmoková, Ali Rezaei-Matehkolaei, Ivana Kuklová, Miroslav Kolařík, Forough Shamsizadeh, Saham Ansari, Maral Gharaghani, Viera Miňovská, Mohammad Javad Najafzadeh, Sadegh Nouripour-Sisakht, Takashi Yaguchi, kamiar zomorodian, Hossein Zarrinfar, and Vit Hubka

Corresponding Author(s): Vit Hubka, Czech Academy of Sciences

Review Timeline:

Submission Date:	May 14, 2021
Editorial Decision:	June 9, 2021
Revision Received:	July 16, 2021
Accepted:	July 26, 2021

Editor: Alexandre Alanio

Reviewer(s): The reviewers have opted to remain anonymous.

Transaction Report:

June 9, 2021

Dr. Vit Hubka
Czech Academy of Sciences
Institute of Microbiology
Prague
Czech Republic

Re: Spectrum00284-21 (Discovery of new Trichophyton members, *T. persicum* and *T. spiraliforme* spp. nov., as a cause of highly inflammatory tinea cases in Iran and Czechia)

Dear Dr. Vit Hubka:

Thank you for submitting your manuscript to Microbiology Spectrum. As you will see the reviewers support publication of a revised paper, and I fully endorse their comments.

Please revise the paper along the lines suggested by the reviewers. When submitting the revised version of your paper, please provide (1) point-by-point responses to the issues raised by the reviewers as file type "Response to Reviewers," not in your cover letter, and (2) a PDF file that indicates the changes from the original submission (by highlighting or underlining the changes) as file type "Marked Up Manuscript - For Review Only". Please use this link to submit your revised manuscript - we strongly recommend that you submit your paper within the next 60 days or reach out to me. Detailed information on submitting your revised paper are below.

Link Not Available

Sincerely,

Alexandre Alanio

Journals Department
Reviewer comments:

Reviewer #1 (Comments for the Author):

The Spectrum manuscript category "Antibacterial and antifungal agents" seems to be incorrectly chosen.

Minor comments:

L177: pesicum > persicum

L214: is "german pointer bitch" correct? "bitch" sounds a kind of 'unpolite'

L223: was also > were also

L225: provenance > origin

L251: species name in italics

L258,260,262,356: CLUSTER into cluster (no reason why to "scream" this by using capital characters

L285,319: check if a spacer needs to be placed between MB and #####

L289,322: check if "metabolic inactive state" needs to be added

L284-340: change "diam" in "diameter"

L375: what happened to *T. indotineae*? Wasn't that the resistant lineage?

L380: this is odd... thus *T. persicum* was already described!?

Reviewer #2 (Comments for the Author):

This is a well written article. I have only a few comments

1. line 256: typo: idiomorp

2. line 271: Four strains were no longer viable for phenotype analyses. Could you please write down between brackets which ones?

3. I think, that the text for figure 6 is misleading, as this tree was constructed with incomplete MLST data for almost all the *Tr. persicum* sp. nov. strains. I suggest to clarify the text to clarify this point. Or: show the tree only with the complete MLST data sets (CCF 6543 for *Tr. persicum* and CCF6259 for *Tr. spiriforme*) and another phylogeny based on ITS with all the strains.

4. Line 366: I think, that it is difficult to confirm reliable species confirmation based on *gapdh*, if the available data is restricted to one sequence for each proposed new species. The same for the identification of *Tr. spiriforme* based on ITS. We do not know the variability of ITS in *Tr. spiriforme*, as there is only one described

5. Table 1: I couldn't find the explanation of ^T. I guess it means ex-type isolates?

Reviewer #3 (Comments for the Author):

Čmoková et al report on the description of two new species of zoophilic dermatophytes which belong to the *T. benhamiae* complex.

In dermatophytes the boundary of species are difficult to draw because the gross of anthropophilic and zoophilic species reproduce clonal. "Clonal lineages that emerged from sexually interacting groups of strains. The original lifecycle of a dermatophytes involves the production of sexual

fruitbodies in the environment. On the mammalian host, cells reproduce mainly clonally; such lineages were termed 'clonal offshoots'. These are molecularly and sometimes also phenotypically a bit different from each other. Dermatophyte species thus consist of clusters of clones, which may or may not interact with mating in nature. However, some of the offshoots can be clinically relevant. This is the case with a virulent, often terbinafine-resistant clone in the *Trichophyton mentagrophytes* complex (Singh et al. 2018). As it is clinically significant to recognize this offshoot, it has been named as *Trichophyton indotineae*. Note that there are more offshoots in the same species complex (Tang et al. 2021), but these don't have a name because they are clinically irrelevant." (de Hoog et al, 3rd edition of the Atlas of clinical fungi).

1. Thus, please summarize in a table the clinical relevance of the two new species compared to the other species in the *T. benhamiae* complex. Is there any relevance in respect to therapy options, source of infection, increased transmission or pathogenicity/virulence?

2. Please include a table of unique qualitative (not quantitative) characters which can be used by the clinician/microbiologist to differentiate both species from each other and more important from phylogenetically close related species in the *T. benhamiae* complex and from *T. verrucosum* and *T. eriothrephon*, like shape of macro and microconidia, pigment production and shape of the colony, physiological features (urease production, T1-T7 agars), ITS signature polymorphisms (because ITS is the standard identification method), Maldi-T of discrimination and animal reservoir.

The authors mention grow parameter, but inaccurate grow parameter (grows slowly) are not a stable character to differentiate species. Or the early formation of spiral hyphae in *T. spiraliforme*. Of *T. spiraliforme* only a single isolate exists! Therefore, the authors can say nothing about the variability of such features.

3. Please indicate the percentage of intraspecific variability within the ITS region for each of the two species.

Microsatellite data are only available for a single isolate per species. These markers are used to structure populations within a species but not to differentiate species. These data can be omitted because they do not add more information to the species description.

4. Kosanke et al (<https://doi.org/10.1016/j.fgb.2018.07.003>) were able to show that separate species in dermatophyte show a certain level of variability in their MAT locus sequence. Please verify this for the two species too.

Staff Comments:

Preparing Revision Guidelines

- Point-by-point responses to the issues raised by the reviewers in a file named "Response to Reviewers," NOT IN YOUR COVER LETTER.
- Upload a compare copy of the manuscript (without figures) as a "Marked-Up Manuscript" file.
- Each figure must be uploaded as a separate file, and any multipanel figures must be assembled into one file.
- Manuscript: A .DOC version of the revised manuscript

- Figures: Editable, high-resolution, individual figure files are required at revision, TIFF or EPS files are preferred

For complete guidelines on revision requirements, please see the Instructions to Authors at [link to page]. **Submissions of a paper that does not conform to Microbiology Spectrum guidelines will delay acceptance of your manuscript.**

Please return the manuscript within 60 days; if you cannot complete the modification within this time period, please contact me. If you do not wish to modify the manuscript and prefer to submit it to another journal, please notify me of your decision immediately so that the manuscript may be formally withdrawn from consideration by Microbiology Spectrum.

If you would like to submit an image for consideration as the Featured Image for an issue, please contact Spectrum staff.

Dear Editorial office, dear reviewers,

thank you for suggestive notes and recommendations. Enclosed please find our manuscript revised according to the recommendations together with responses (**R:**) to comments (**C:**) in a point by point manner.

Changes are also highlighted by yellow color in the file "Marked up Manuscript".

REVIEWER #1 (COMMENTS FOR THE AUTHOR):

Minor comments:

C: L177: pesicum > persicum

R: Corrected.

C: L214: is "german pointer bitch" correct? "bitch" sounds a kind of 'unpolite'

R: It is correct but we found another designation for this breed "German Shorthaired Pointer" and replaced the original name with this new.

C: L223: was also > were also

R: Rephrased.

C: L225: provenance > origin

R: Replaced as suggested.

C: L251: species name in italics

R: Corrected.

C: L258,260,262,356: CLUSTER into cluster (no reason why to "scream" this by using capital characters

R: Corrected as suggested.

C: L285,319: check if a spacer needs to be placed between MB and #####

R: We added a space.

C: L289,322: check if "metabolic inactive state" needs to be added

R: It is not needed as we deposited dried herbarium specimens (mentioned in Methods). This statement is only needed if cultures (lyophilized or stored in liquid nitrogen) are used as holotypes.

C: L284-340: change "diam" in "diameter"

R: Changed as suggested.

C: L375: what happened to *T. indotineae*? Wasn't that the resistant lineage?

R: We replaced "*T. mentagrophytes* and *T. rubrum*" with "*T. mentagrophytes* and *T. rubrum* complexes". *Trichophyton indotineae* is a member of *T. mentagrophytes* complex and it is still designated by many authors as "*T. mentagrophytes* ITS genotype VIII". Although this species was validly described, many researchers have some doubts about its taxonomic status and consider it a lineage within *T. mentagrophytes*.

C: L380: this is odd... thus *T. persicum* was already described!?

R: No, but several *T. persicum* strains have been tested in previous studies under the name *T. benhamiae* and these strains were described in this study as *T. persicum* and their identification confirmed using sequences deposited in GenBank. We slightly rephrased the text to avoid misinterpretation.

REVIEWER #2 (COMMENTS FOR THE AUTHOR):

This is a well written article. I have only a few comments

C: 1. line 256: typo: idiomorp

R: Corrected.

C: 2. line 271: Four strains were no longer viable for phenotype analyses. Could you please write down between brackets which ones?

R: The strain numbers were added.

C: 3. I think, that the text for figure 6 is misleading, as this tree was constructed with incomplete MLST data for almost all the *Tr. persicum* sp. nov. strains. I suggest to clarify the text to clarify this point. Or: show the tree only with the complete MLST data sets (CCF 6543 for *Tr. persicum* and CCF 6259 for *Tr. spiraliforme*) and another phylogeny based on ITS with all the strains.

R: We added both, phylogeny with all samples based on ITS rDNA region and also phylogeny which is based only on the isolates with complete MLST data (Supplementary Fig. S1 and Fig. S2). The topology of the these trees is similar with some changes in deeper nodes which gained minimal support values in all phylogenies. Maximum likelihood method, when correct partitioning scheme is used (as we did), gives very good results in case of incomplete data and this is a common practice in phylogenetic studies.

C: 4. Line 366: I think, that it is difficult to confirm reliable species confirmation based on *gapdh*, if the available data is restricted to one sequence for each proposed new species. The same for the identification of *Tr. spiraliforme* based on ITS. We do not know the variability of ITS in *Tr. spiraliforme*, as there is only one described

R: Yes and no. The intraspecific variability within *T. benhamiae* complex species is extremely low. In some species, there is no variability in the four sequenced loci despite the sequences were generated across many strains from different hosts and countries. And there are only few exceptions like *T. africanum* and *T. erinacei* (unpublished data) with some intraspecific variability higher than few substitutions. For instance, in the *gapdh* gene, there are 12 substitutions between *T. spiraliforme* and *T. persicum* ex-types. And this is high number which exceeds intraspecific variability in any other dermatophyte species with available data for this gene. In this regard, it is very unlikely that these 12 substitutions represent an intraspecific variability. On the other hand, it is true that some of these substitutions may not be species-specific and the final number will be lower when more strains will be analyzed. The real number of species-specific substitutions needs to be definitely verified in future studies. We have modified and mitigated our statements in the Results in this way. Thank you for this comment.

C: 5. Table 1: I couldn't find the explanation of ^T. I guess it means ex-type isolates?

R: Yes, superscript "T" designates the ex-type strain. We added explanation into the footnote of Table 1.

REVIEWER #3 (COMMENTS FOR THE AUTHOR):

C: Čmoková et al report on the description of two new species of zoophilic dermatophytes which belong to the *T. benhamiae* complex.

In dermatophytes the boundary of species are difficult to draw because the gross of anthropophilic and zoophilic species reproduce clonal. "Clonal lineages that emerged from sexually interacting groups of strains. The original lifecycle of a dermatophytes involves the production of sexual fruitbodies in the environment. On the mammalian host, cells reproduce

mainly clonally; such lineages were termed 'clonal offshoots'. These are molecularly and sometimes also phenotypically a bit different from each other. Dermatophyte species thus consist of clusters of clones, which may or may not interact with mating in nature. However, some of the offshoots can be clinically relevant. This is the case with a virulent, often terbinafine-resistant clone in the *Trichophyton mentagrophytes* complex (Singh et al. 2018). As it is clinically significant to recognize this offshoot, it has been named as *Trichophyton indotineae*. Note that there are more offshoots in the same species complex (Tang et al. 2021), but these don't have a name because they are clinically irrelevant." (de Hoog et al, 3rd edition of the Atlas of clinical fungi).

1. Thus, please summarize in a table the clinical relevance of the two new species compared to the other species in the *T. benhamiae* complex. Is there any relevance in respect to therapy options, source of infection, increased transmission or pathogenicity/virulence?

R: Our study is taxonomic research. We agree that species concept in dermatophytes is still evolving and there are different views on how species should be defined. And it is true that the same species concept cannot be applied on all species. Among anthropophilic and zoophilic dermatophytes, there are many strictly clonal species, and some species with higher degree of variability (e.g. *T. mentagrophytes*), showing signs of recombination within the population similar to geophilic dermatophytes. In these cases, different concept should be applied.

But on the other hand, we do not support the idea that taxonomic conclusions should be based on the clinical relevance of species. In this regards, we do not agree that *Trichophyton indotineae*, which is presented to us as a model, is a good example of taxonomic work. With all respect to authors of this article, the species concept was only based on misleading presentation of ITS phylogeny, terbinafine resistance and clinical manifestation. The last two mentioned characteristics are not taxonomically informative. It has already been shown by some studies (e.g. Kong et al. 2021, doi: 10.1128/AAC.00056-21) that the terbinafine resistance is only restricted to ca 50% of *T. indotineae* strains. And clinical manifestation of *T. indotineae* infections is not only typical for *T. indotineae*. The authors, who described *T. indotineae*, examined also some morphological and physiological characteristics which are however not different from the closely related *T. mentagrophytes/interdigitale*. For these reasons, this study is not an example we want to follow.

Regarding the other tentative offshoots in the *T. mentagrophytes* lineage which have not been proposed as separate species by Tang. et al. (2021): It is very questionable, that the other lineages are clonal offshoots. Based on the single gene trees presented by Tang. et al. (2021), it can be asserted with high probability that there is an intense recombination within *T. mentagrophytes* lineage and this is further supported by the presence of both mating-type gene idiomorphs in the population of this species. So in our opinion, these lineages do not fit the definition of "clonal offshoots" as they are not genetically isolated from other lineages and rather form a broad sexual species.

The species proposed in the current study cannot be classified into any of the currently existing species in the *T. benhamiae* complex and we used independent genetic markers to confirm it. And we also supported the molecular data by observations of some differences in the morphological characters. We are aware of limitations of our study as discussed below and we cannot be 100% sure if the future will bring some correction of species concept in dermatophytes as taxonomy is an evolving opinion.

Both species are clinically relevant as they cause highly inflammatory tinea infections suggestive of zoonotic dermatophytosis. We discussed treatment options and probable source of infections in the section Discussion. The treatment is similar to other infections caused by *T. benhamiae* complex species and zoonotic tinea infections in general. In this aspect, we have nothing to compare across zoophilic dermatophytes and the supposed Table would be superfluous. We do not have data to say something about „increased

transmission or pathogenicity/virulence” and such data does not exist for the majority of dermatophyte species.

C: 2. Please include a table of unique qualitative (not quantitative) characters which can be used by the clinician/microbiologist to differentiate both species from each other and more important from phylogenetically close related species in the *T. benhamiae* complex and from *T. verrucosum* and *T. eriothrephon*, like shape of macro and microconidia, pigment production and shape of the colony, physiological features (urease production, T1-T7 agars), ITS signature polymorphisms (because ITS is the standard identification method), MALDI-ToF discrimination and animal reservoir.

R: We summarized selected characteristics in the new scheme - Figure 9. We also added Supplementary Table 1 summarizing all stable substitutions across species from *T. benhamiae* clade. ITS signature polymorphisms are summarized in Table S1. Testing on Trichophyton agars was not performed because the data are not available across the *T. benhamiae* complex. Similarly, MALDI-TOF spectrometry was not performed as responsible comparison of spectra across the *T. benhamiae* complex would require a separate study and several months of work.

C: The authors mention grow parameter, but inaccurate grow parameter (grows slowly) are not a stable character to differentiate species. Or the early formation of spiral hyphae in *T. spiraliforme*. Of *T. spiraliforme* only a single isolate exists! Therefore, the authors can say nothing about the variability of such features.

R: We are aware of these limitations and they are clearly mentioned in the text. Growth parameters are given in mm in the section Taxonomy. We only used designation “slow” or “rapid” growth in the differential diagnosis from other species and reader can find the exact dimensions in the species descriptions. The fact that *T. persicum* grows slowly in comparison with related species is not only based on the data from the ex-type strain, but also from several other strains which were identified by clinicians as *T. verrucosum* because of their slow growth and sporulation. The growth parameters are relatively stable across species of the *T. benhamiae* complex. In our previous study (Čmoková et al. 2020, Fungal Divers 104:333–387), we examined a large number of strains of various species and there was a very low number of strains having significantly different colony sizes. The relatively high variability in morphology of colonies and sometimes also in micromorphology is a common problem across many dermatophyte species. This is not only restricted to related species but sometimes we observe very similar morphology in unrelated species having different ecology and clear support by molecular markers (e.g., *T. mentagrophytes* – *T. quinckeanum* – *T. erinacei*). The size and shape of spores usually belongs to the most stable characters. However, there is usually an overlap between species.

We do not claim anywhere in the text that we know something about intraspecific variability and thus, no exclamation is needed. We tried to find unique morphological characters with our knowledge of morphology of the other species in the complex. And we believe that we found some. For instance the growth parameters on some media: if you compare (Figure 9) dimensions of colonies of the new species with the most closely related species, *T. europaeum* and *T. japonicum*, they are not even overlapping on some media, although the data are available for many *T. europaeum* and *T. japonicum* strains from our previous study. We believe that these characters will prove to be taxonomically significant and will be useful for identification.

The new species proposed here are phylogenetically unique and they also show differences from related species in characters which are stable in those species. It is highly improbable that this is just coincidence and that we only isolated some genetically exceptional strains which have at the same time unique phenotype. That would be too many coincidences at

once. On the other hand, we clearly stated that these observations need to be confirmed in future when more strains will be available.

It is a very common situation in today's taxonomy that species are described on the basis of one or several isolates because it is not possible to obtain new isolates within a reasonable time frame, or it requires a significant effort with an uncertain outcome. This is also our case. *Trichophyton spiraliforme* represents an unique finding among 10 000 of strains which we have examined in the last decade. In addition, we did not find any similar strains using BLAST similarity search. This does not necessarily mean that it is an extremely rare species because it is probably an animal pathogen. And unfortunately, very low number of sequences are available from animal dermatophytoses caused by "*T. benhamiae*".

C: 3. Please indicate the percentage of intraspecific variability within the ITS region for each of the two species.

R: The information about intraspecific variability among *T. persicum* isolates is mentioned in the section Results. We cannot include this information for *T. spiraliforme* because we have only one strain. Stable positions differentiation species from *T. benhamiae* clade are listed in Table S1.

C: Microsatellite data are only available for a single isolate per species. These markers are used to structure populations within a species but not to differentiate species. These data can be omitted because they do not add more information to the species description.

R: We agree that microsatellite data are used for population genetic studies as we mentioned in the text. But in dermatophytes, they are also useful for classical taxonomy as there is a lack of sufficiently variable markers. The analysis of microsatellites shown in this study is very valuable even with the limited number of strains because the data clearly show that *T. persicum* and *T. spiraliforme* do not belong to any population of existing species. In our dataset, there are more than 300 strains from different countries and hosts, and the isolates of the same species form clusters which correspond with multi-gene phylogeny. And in agreement with phylogeny, *T. persicum* and *T. spiraliforme* form a separate branch with a deep node. This shows that these species are distant from currently known species and from each other. So this is another confirmation by independent markers (in addition to MLST data) that these isolates are unique and deserve to be recognized as separate species in line with the current species concept in the *T. benhamiae* complex.

C: 4. Kosanke et al (<https://doi.org/10.1016/j.fgb.2018.07.003>) were able to show that separate species in dermatophyte show a certain level of variability in their MAT locus sequence. Please verify this for the two species too.

R: In our study, we used primers TmHMG3S and TmHMG3R for amplification of MAT1-2-1 gene. These primers amplify only very small fragment of about 100 bp and are used for routine confirmation of presence of this MAT gene idiomorph on the electroforetogram. Although we also sequenced PCR products for verification of specificity, this region is very conserved in all *Trichophyton* species and does not enable detailed comparison between species.

Following this review, we used another primer combinations to amplify longer fragment in our new species and we were only successful in *T. persicum*. Based on this sequence, we can say, that MAT1-2-1 gene of *T. persicum* is more related to *T. erinacei* than to *T. benhamiae* var. *benhamiae* (sequences from study of Kano et al. 2012, e.g. AB570252 and AB542198) and *T. europaeum* (genomic data from strain CBS 112371 and our unpublished data). Detailed phylogenetic analysis across species diversity of *Trichophyton* is not possible because Kosanke et al. (2018) deposited only five MAT gene sequences for *Trichophyton* species and none of them belong to *T. benhamiae*. Data from other studies comprise very different regions of MAT1-2-1 genes which cannot be used in the same alignment because of

their small overlap. Thus our knowledge on evolution of MAT genes in *Trichophyton* is still rather superficial and certainly deserved further investigations. We decided to not include our incomplete data in the present manuscript because this information would be rather misleading. We aim to include analysis of MAT genes in some of the future studies on *T. benhamiae* complex which are in preparation.

Kind regards,

Vit Hubka, Corresponding author

Faculty of Science, Charles University
Benatska 2, 12801 Prague 2
Czech Republic

Email: vit.hubka@gmail.com

Phone: (+420) 739663218

July 26, 2021

Dr. Vit Hubka
Czech Academy of Sciences
Institute of Microbiology
Prague
Czech Republic

Re: Spectrum00284-21R1 (Discovery of new Trichophyton members, *T. persicum* and *T. spiraliforme* spp. nov., as a cause of highly inflammatory tinea cases in Iran and Czechia)

Dear Dr. Vit Hubka:

I would like to thank you for your efforts in responding to reviewer's comments.

Your manuscript has been accepted, and I am forwarding it to the ASM Journals Department for publication. You will be notified when your proofs are ready to be viewed.

Sincerely,

Alexandre Alanio
Editor, Microbiology Spectrum

Journals Department
Fig. S1, Fig. S2, Table S1: Accept